# Sound Absorption Performance and Mechanical Properties of the 3D-Printed Bio-Degradable Panels

**DOI:** 10.3390/polym15183695

**Published:** 2023-09-07

**Authors:** Sebastian-Marian Zaharia, Mihai Alin Pop, Mihaela Cosnita, Cătălin Croitoru, Simona Matei, Cosmin Spîrchez

**Affiliations:** 1Department of Manufacturing Engineering, Transilvania University of Brasov, 500036 Brasov, Romania; zaharia_sebastian@unitbv.ro; 2Department of Materials Science, Transilvania University of Brasov, 500036 Brasov, Romania; simona.matei@unitbv.ro; 3Department of Product Design, Mechatronics and Environment, Transilvania University of Brasov, 500036 Brasov, Romania; mihaela.cosnita@unitbv.ro; 4Materials Engineering and Welding Department, Transilvania University of Brasov, 500036 Brasov, Romania; c.croitoru@unitbv.ro; 5Wood Processing and Design Wooden Product Department, Transilvania University of Brasov, 500036 Brasov, Romania; cosmin.spirchez@unitbv.ro

**Keywords:** 3D printing, acoustic properties, mechanical properties, bio-degradable panels

## Abstract

The 3D printing process allows complex structures to be obtained with low environmental impact using biodegradable materials. This work aims to develop and acoustically characterize 3D-printed panels using three types of materials, each manufactured at five infill densities (20%, 40%, 60%, 80% and 100%) with three internal configurations based on circular, triangular, and corrugated profiles. The highest absorption coefficient values (α = 0.93) were obtained from the acoustic tests for the polylactic acid material with ground birch wood particles in the triangular configuration with an infill density of 40%. The triangular profile showed the best acoustic performance for the three types of materials analysed and, from the point of view of the mechanical tests, it was highlighted that the same triangular configuration presented the highest resistance both to compression (40 MPa) and to three-point bending (50 MPa). The 40% and 60% infill density gave the highest absorption coefficient values regardless of the material analyzed. The mechanical tests for compression and three-point bending showed higher strength values for samples manufactured from simple polylactic acid filament compared to samples manufactured from ground wood particles. The standard defects of 3D printing and the failure modes of the interior configurations of the 3D-printed samples could be observed from the microscopic analysis of the panels. Based on the acoustic results and the determined mechanical properties, one application area for these types of 3D-printed panels could be the automotive and aerospace industries.

## 1. Introduction

In recent times, with the development of modern industry and air and road traffic, noise has become one of the countless factors affecting human health and the environment worldwide. Noise has become a major problem that causes harmful effects on human health, and combating it requires urgently finding the best solutions through increasingly restrictive regulations and legislation [1,2]. Lately, the improvement of living conditions by regulating noise in the fields of aviation (by reducing the interior noise of passenger aircraft), railways, automobiles and buildings is a requirement forced on industrial companies [3].

In modern industry, noise is one of the most urgent issues to be addressed, researched, and reduced using sound-absorbing materials. For sound reduction, structural design and testing of sound absorption and sound transmission loss of industrial products is challenging and of great interest [4,5,6]. Nowadays, various types of natural materials [7,8,9], composite materials [10,11,12,13] and composite sandwich structures [14,15,16,17] are acoustically researched because of their low production cost and often environmentally friendly compositions. The application of 3D-printed materials in the aerospace and automotive industry involves higher expenses that are especially justified for prototypes and customized components. In general, the materials used with the FFF process have much lower costs compared to classic technologies (plastic mass injection of components) or additive technologies such as SLS. When sound reaches a barrier, depending on the sound absorption performance of the material, six phenomena can occur: absorption, transmission, reflection, refraction, scattering and diffraction [18].

The main coefficients that evaluate the acoustic performance of materials are the sound absorption coefficient and sound transmission loss [19]. The sound absorption coefficient reflects the ratio between reflected and incident sound intensity (W/m^2^). The proper physical quantity is sound intensity, and it varies as a function of the frequency [20,21] and angle [22] at which a sound or sound wave reaches the material under test. Sound transmission loss is the ratio of the transmitted sound energy to the amount of sound energy remaining on the incident side of the material under test. For the design and manufacture of industrial products requiring high acoustic performance, material selection and acoustic testing of selected materials are two very important activities. From the reviewed studies [23,24,25,26,27], the most important factor influencing the acoustic performance of materials is the porosity value.

Currently, additive manufacturing processes using plastics, composites and metallics are considered the most developed and researched processes, which, in the near future, could play a role in traditional manufacturing processes in many industries (aerospace, automotive, marine engineering, medical and many others). Thus, the most used manufacturing process for the development of porous structures subject to acoustic performance studies is fused filament fabrication (FFF) or 3D printing. In a recent study [28] the acoustic performance of a 3D-printed biodegradable material with bubble holes of different sizes was analysed. The results showed improved performance for different sizes of spherical bubbles and different types of hole patterns in the low-frequency range up to 1000 Hz.

Sailesh et al. [29] used polylactic acid (PLA) granules to manufacture perforated samples with different cross-sections via the FFF process for the purpose of determining acoustic indicators. It was found that the maximum values of the sound absorption coefficient are obtained in the range 500–1000 Hz. Other studies [30,31] have shown a significant influence of pore shape, volume ratio, material thickness, and air gap size of 3D-printed acrylonitrile butadiene styrene filament (ABS) structures on acoustic performance. Zielinski et al. [32] proposed and acoustically analyzed a sound-absorbing material, whose acoustic performance resulted not only from the designed pore network but also from the microporosity of the material used during the additive manufacturing process. In this way, a dual porosity material was successfully designed, modelled, and manufactured based on the imperfections of the materials in the manufacturing process to obtain high acoustic performance. Errico et al. [33] studied the vibro-acoustic behaviour of 3D-printed panels using periodic structure theory over a wide frequency range (1000–10,000 Hz). Boulvert et al. [34] 3D printed micro-lattices to study the manufacturing accuracy and prediction of its absorption characteristics using the Johnson–Champoux–Allard–Lafarge (JCAL) model in order to draw conclusions about the defects of the 3D printing process. Thus, the accuracy was affected by the nozzle cross-section shape, which was about 8%, but there were also deviations attributed to thermal shrinkage and micro-grooves of the nozzle. Reentrant auxetic structures are used in various noise reduction applications (in the automotive and aerospace fields). Hence, such cellular structures were modelled and fabricated [35] using keratin-reinforced polylactic acid, which were 3D printed and were able to reduce noise as they reduce the voids and airflow.

With the development of additive manufacturing processes, there is a trend towards the development of acoustically absorbent metamaterials capable of controlling, guiding, and manipulating low- and medium-frequency acoustic waves. Gao and Hou [36] researched the sound absorption coefficient of a 3D-printed polylactic acid microhelix metamaterial. Their research found that the sound absorption coefficient values improve as the gap between the micro-helices increases.

Other studies [37,38] have researched the acoustic properties of multi-layer perforated panels manufactured by additive manufacturing processes (selective laser sintering and stereolithography). The SLA additive process was used to manufacture porous polycarbonate samples to study the effects of hole angle and air gap on sound absorption [39]. The results revealed that increasing the angle of inclination of the pores reduced the value of the sound absorption coefficient when the porosity was kept constant. Jiang et al. [40] researched the feasibility of manufacturing materials by the MultiJet 3D printing process and determined the sound absorption capability of materials with different geometrical parameters (porosity, hole diameter, sample thickness and effect of aspect ratio). The results indicated that the maximum absorption coefficients ranged from 0.24 to 0.99, and for the samples to have good sound absorption performance, they were tested at high frequencies from 4800 Hz to 6400 Hz.

In the research [41], the sound absorption of a micro-perforated panel (MPP) manufactured by the 3D printing process was investigated, and the sound absorption coefficient of a 3D-printed MPP layer supported with a porous material was measured and theoretically predicted using the transfer matrix method (TMM). Moreover, another study available in the literature argued for the use of microstructures with relatively simple pores, such as parallel, identical, and inclined apertures, for the analysis of broadband sound absorption through analytical and additively manufactured models [42]. Sekar et al. [43] focused on the study of micro-perforated panels (MPP) made of polylactic acid (PLA) reinforced with wood fibres fabricated using the FFF process. Acoustic test results indicated that changing the perforation volume affects the acoustic absorption of the MPP. MPP with a thickness of 2 mm and a perforation diameter of 0.2 mm presents a maximum sound absorption coefficient value of 0.93 at a frequency of 2173 Hz.

Currently, numerous studies are aimed at developing sustainable and high-performance materials for manufacturing perforated panels from natural resources. However, obtaining complex holes with different geometries from different materials is an issue that requires detailed acoustic studies and analysis. Therefore, in this study, the acoustic behaviour of perforated panels consisting of three types of materials (PLA with a mixture of 40% ground coconut wood particles, PLA with a mixture of 40% ground birch wood particles and plain PLA) manufactured via the FFF process and were tested using plane rolling waves with an incident perpendicular to the tested surface (tube method). Acoustic analysis of the panels was carried out for five values of infill density (20%, 40%, 60%, 80% and 100%), with three internal configurations (circular profile, triangular profile and corrugated profile). The panel configurations with the best acoustic performance were manufactured via the FFF process and were mechanically tested (compression and bending). Figure 1 briefly describes the organisation of this study.

## 2. Materials and Methods

### 2.1. Design of the Panels

The sample design was carried out in the SolidWorks 2016 software system considering the standards specific to acoustic testing [44,45] and compression and bending testing of sandwich panels [46,47]. The 3D-printed samples for acoustic testing have the following dimensions (Table 1): the upper and lower parts have diameters of 50 mm, thicknesses of 8 mm and holes of 4.2 mm required for the assembly of the two parts. These dimensions are in accordance with the standard as well as with the technical characteristics of the impedance tube. The stages of the acoustic tests and the three-dimensional models of the samples are presented in Table 1 as follows: in the first stage the double panels were tested; the second stage consisted of testing the single panels; the third stage was dedicated to testing the single panels on which holes were drilled by means of a 3D-printed template; and in the last stage, the panels were 3D printed with a triangular internal configuration and a rhombic profile. A side of 1.8 mm was chosen for the holes.

The aim was to compare the way the shape of the gaps influences the sound-absorbing properties. The modes of reflection and absorption of sound waves were a circular profile (gaps that are not interconnected), an interconnected corrugated one (interconnected ovate gaps) and a triangular one (walls with right angles). Those three cases, in our opinion, can cover a large area of cases encountered in practice.

The dimensions of the samples tested at compression were a width of 50 mm and a thickness of 15 mm (Table 2). The samples for the three-point bending tests were manufactured via the FFF process with the following dimensions: length 150 mm, width 20 cm, and thickness 15 mm (Table 2). For the acoustic testing of the 3D-printed specimens, the (ASTM) E1050 [44] and ISO 10534-2 [45] standards were followed.

### 2.2. Materials Properties

The samples for the experimental tests (acoustic and mechanical) were manufactured from three types of materials: standard PLA [45], PLA with a mixture of 40% ground coconut wood particles [48] and PLA with a mixture of 40% ground birch wood particles. According to the information provided by the filament manufacturer, the particle size of birch and coconut is between 70 and 140 µm [48].

PLA material is one of the most used thermoplastic polymeric materials suitable for the FFF process. PLA has the following advantages [49,50,51]: it is a low-cost biodegradable polymer; it is very easy to manufacture via the FFF process; it is the best material in terms of dimensional accuracy (it does not undergo deformation during the FFF process nor after cooling); it has good adhesion to the bed plate but also between the extruded layers of material and during the manufacturing process, unpleasant smells are not emitted. PLA reinforced with 40% ground wood particles as a filament very easy to manufacture via the FFF process as it uses the PLA as base (matrix) material and has several important advantages [48,52]: it is a biodegradable material that is easy to manufacture; it has a wood-like smell and appearance; it has good adhesion in the first layer of manufacturing.

### 2.3. Manufacturing Process of the 3D-Printed Panels

The three types of filaments (PLA coconut, PLA birch and PLA) with different mechanical properties were used for the FFF manufacture of 3D-printed bio-degradable perforated panels. The 3D-printed panels were manufactured via the FFF process with the CreatBot DX-3D double-nozzle printer (Henan Suwei Electronic Technology Co., Ltd., Zhengzhou, China). The manufacturing parameters settings were selected according to the filament type and were controlled via the 3D printing slicing software CreatBot V6.5.2. The main 3D printing parameters of printed panels with the 3 types of filaments are presented in Table 3.

### 2.4. Acoustic Testing

The sound absorption behaviour of samples manufactured via the FFF process was researched using a Holmarc HO-ED-A-03 acoustic impedance tube (Holmarc Opto-Mechatronics Ltd. Kochi, India), which was equipped with the following: hollow tubes, two pairs of microphones, sample holders, a data acquisition system and measurement software. The impedance system contains an anodised aluminium tube with an internal diameter of 50 mm, which can measure in the frequency range 500–3150 Hz.

In this study, the frequency dependencies of the sound absorption coefficient (α) and the sound transmission loss (STL) of 3D-printed samples were investigated by the transfer function method according to the current standards [44,45]. Figure 2a shows the components of the impedance tube system used for acoustic testing. Figure 2b shows the two schematic configurations of the impedance tube system by means of which the acoustic performance of 3D-printed samples can be determined: for the determination of the sound absorption coefficient, the system also contains the anechoic termination component, and for the sound transmission loss, this anechoic termination component has been removed. For each tested sample, the geometric parameters of the samples (50 mm), microphone spacing (30 mm), temperature and humidity recorded at each current test were entered.

### 2.5. Mechanical Testing

Mechanical tests were carried out using the W-150 S universal testing machine (Jinan Testing Equipment IE Corporation, Jinan, China). Compression testing (Figure 3a) was carried out at a loading speed of 5 mm/min according to the current standards [46,47] and highlights the behaviour and response of the 3D-printed bio-degradable panels under a compressive load by measuring the fundamental characteristics (compressive strength and load–displacement curves). The 3D-printed samples were tested in three-point bending according to the requirements of the standards [46,47] in order to determine the key characteristics (bending strength and load–displacement curves). Five samples were tested according to the standards for testing sandwich structures for both types of tests (compression and three-point bending).

### 2.6. Microscopic Analysis

Electronic Nikon Eclipse MA 100 microscopes (Nikon Corp., Tokyo, Japan) were used to examine the condition of the 3D-printed structures after compression testing in order to detect typical defects and specific failure modes.

## 3. Results and Discussion

### 3.1. The Effect of Filling Density on the Acoustic Performance of 3D-Printed Double Panels

Double panels have been designed as consisting of two half-panels (demountable assembly), dedicated to repairs when necessary (damage of only one surface) thus achieving a reduction in manufacturing time and costs associated with their manufacture by replacing only the damaged half-panel. Ultimately, the panels are designed and manufactured in mirror image to each other, thus obtaining as large internal voids as possible, which causes a decrease in the mass of the panel and implies an economy of material and reduced costs.

Analysing all the experimental data obtained for the 3D-printed panels, graphs are drawn for all the samples to observe the differences and variations of the sound absorption coefficient and loss transmission as a function of frequency.

In the first stage of the study, a design was made of the three types of filaments (PLA coconut, PLA birch, and PLA) and with the five types of infill density (20%, 40%, 60%, 80% and 100%) in order to characterize the acoustic performance of those three types of materials. Since the sound absorption coefficient is a ratio of reflected to incident sound energy, a value of 0 indicates complete reflection with no absorption [53]. On the other hand, the value of 1 of the absorption coefficients shows that all the sound energy is absorbed without reflection [53].

For the first acoustic tests (Figure 4), double panel assemblies were used, where the fill density varied as follows: 20%, 40%, 60%, 80% and 100%. These fill density variations were analysed on three types of filaments (PLA–birch, PLA–coconut and PLA) using the three profiles (circular, triangular, and corrugated/undulated). Thus, a total of 45 acoustic tests were carried out for the 3D-printed double panels, where the two most important parameters were determined: sound absorption coefficient (SAC, *α*) and the sound transmission loss (STL).

For each type of material, the curves for the highest values of sound absorption coefficient (α) were plotted. The markings on the figures are as follows: 20%, 40%, 60%, 80% and 100%. These represent the infill densities for circular, triangular, and corrugated; they are the internal configurations used and PLA–birch, PLA–coconut and PLA represent the materials from which the samples are made. In the case of the double panels, manufactured via the FFF process, the following can be noticed: for the PLA–birch filament, the highest value of sound absorption coefficient (α = 0.4) was for the 100% corrugated sample at low frequencies (500 Hz), and the highest value for sound transmission loss (STL = 63 dB) was for 60% corrugated; for the PLA–coconut filament, the highest value of sound absorption coefficient (α = 0. 33) was for the 40% triangular sample at low frequencies (500 Hz), and the highest value for sound transmission loss (STL = 64 dB) was for 80% triangular; for the PLA filament, the highest value for sound absorption coefficient (α = 0.36) was for the 80% triangular sample at low frequencies (500 Hz), and the highest value for sound transmission loss (STL = 62 dB) was for 20% circular.

It can therefore be concluded that the variation of the filling density influences the acoustic performance, and samples with lower filling density result in a higher absorption coefficient. Decreasing the filling density improves the passage of acoustic waves entering the panel, and the air inside the voids can now move easily, which increases the viscous friction, causing a loss of acoustic wave energy and, thus, the sound is absorbed more efficiently [54].

Also, acoustic analyses cannot clearly establish a filling density that shows superior performance compared to the others for all filament types analysed. This finding is based on the typical defects (voids, inter-track voids between the layers) that occur in the parts manufactured via the FFF process [55,56]. In contrast, at this stage, when analysing double panels, it can be concluded that the PLA–birch filament showed the highest sound absorption coefficient (α = 0.4). At the same time, the sound transmission loss presented the highest value (STL = 64 dB) in the PLA–birch–triangular configuration at 80% filling density. On the other hand, when analysed in terms of internal configurations, it can be deduced that the triangular configuration type shows the highest values of absorption coefficient and sound transmission loss in two of the three configurations. The triangular configuration shows higher absorption due to the cell walls interacting with the ultrasonic wave along the in-plane direction [57].

The highest sound absorption coefficient was at the frequency of 500 Hz, namely 0.4 for the PLA–birch–triangular configuration at 80% infill density. The results obtained indicate that approximately 60% of the sound energy is reflected without penetrating the surface of the test specimen. From the results of the acoustic tests, the highest sound absorption coefficient is found at low frequencies (500 Hz), which indicates that the specimen acts as a deflector that transfers the vibro-acoustic energy, providing some damping [53].

Also, the difference between the highest and lowest values of the absorption coefficient was 0.07, which indicates a low absorption of the three types of materials manufactured via the FFF process. The sound absorption coefficient represents an important result in the acoustic tests of the materials (PLA–coconut, PLA–birch, and PLA), but it is not the only significant result. Increased importance is also given to the sound transmission loss parameter for 3D-printed specimens, through which a characterization of a specimen’s ability to block sound can be performed.

In the case of these double panels, the alpha coefficient decreases with increasing frequency and the LTR increases very slightly with increasing test frequency.

### 3.2. Mechanical Performance of 3D-Printed Double Panels

From the results of the acoustic tests, two basic criteria were set for determining the samples to be subjected to mechanical tests: the samples with the highest absorption coefficient depending on the material and the samples with the highest absorption coefficient depending on the profile configuration. Thus, for the three-point compression and bending mechanical testing, four sample configurations were manufactured and tested: 60% circular *PLA*–birch, 100% corrugated *PLA*–birch, 40% triangular–*PLA*–coconut, 80% triangular–*PLA*. Two configurations (80% triangular–*PLA* and 100% corrugated *PLA*–birch) showed the best acoustic performance on both criteria analysed. Five samples were tested for each type of test (compression and three-point bending).

The results of the compression tests were presented below as load–displacement characteristic curves (Figure 5a) for each type of 3D-printed structure. Analysing Figure 5a and considering the numerical results provided by the testing machine in the test report, the maximum breaking force of the 80% triangular–*PLA* sample reached the maximum value of 90 kN at a displacement of 2.75 mm during the compression process. On the other hand, the 40% triangular–*PLA*–coconut sample showed the lowest maximum compression force of about 23 kN. For the five samples tested, in each configuration, the average values of compressive strength and compressive modulus of elasticity were calculated and plotted (Figure 5b). The average compressive strength value of the 80% triangular–*PLA* samples showed the highest value of about 40 MPa. From the analysis of the tested samples, it can be deduced that the parts with a mixture of 40% ground birch and coconut wood particles show close values of compressive strength, but not exceeding 13 MPa.

This was due to the lower mechanical performance of the ground wood introduced in this type of filament, i.e., the weaker adhesion between the matrix (PLA) and the ground wood. Another important aspect is the choice of the configuration type for the 3D-printed samples. It is observed that the triangular profile absorbs the compressive stress much better compared to the other two profiles (circular and corrugated) due to the high-stiffness inner walls.

Following the three-point bending tests of the 3D-printed samples, the characteristic curves of the four types of samples and the bending performance (bending strength and modulus of elasticity) were determined. In this case, the 80% triangular–PLA samples also showed the best three-point bending performance. This is due to the specific bending behaviour: during loading, both tensile (on the lower shell) and compressive (on the upper shell) stresses occur simultaneously in the 3D-printed sample, and the core, in this case, is shear stressed. Core shear occurs quite late because the triangular configuration has very strong walls that maintain the structure during loading. The tests showed that the maximum breaking force of the 80% triangular–PLA sample reached a maximum value of 1.58 kN at a displacement of 6.3 mm during the stretching process (Figure 6a). The maximum value of the bending strength was obtained for the 80% triangular–PLA samples. Analysing the other types of samples made of ground wood, it can be observed that the highest bending strength (Figure 6b) is shown by the 100% corrugated PLA–birch samples due to the 100% filling density, which was confirmed in other recent studies [58,59].

### 3.3. Microscopic Analysis of 3D-Printed Double Panels

For the microscopic analysis of the samples, the 3D-printed double panels (Figure 7) were cross-sectioned, embedded into resin, and polished using a 1 μm grit and 0.5 μm Al_2_O_3_ suspension.

The microscopic analysis of 3D-printed double panels was carried out to exemplify the deformations of the internal configurations and at the same time to verify the deposition behaviour of the extruded material at different filling densities. The samples that were microscopically analysed were subjected to plane compression tests. In Figure 7a, the sample underwent a deformation in the middle of the part, followed by a crack with propagation in the middle of the part. Interlayer voids are also observed on the left and right sides. In Figure 7b, the following can be observed: uniform distribution of birch ground wood particles, interlayer crack defects, and a reduced number of defects when adding layers of extruded material.

Following the compression tests (Figure 7c), the walls of the 40% triangular–PLA–coconut samples were deformed by lateral buckling. At the same time, a closer analysis of these samples (Figure 7c) reveals end layer voids, defects with different configurations (triangular and parallelepipedal) and porosity defects (voids) between the successive beads, also found in other studies and specific to the FFF process [60,61,62].

Analysing the 80% triangular–*PLA* sample (Figure 7d), inter-layer voids and porosity defects were identified, which led to higher performance of acoustically tested samples [32].

### 3.4. Acoustic Analysis of 3D-Printed Single Panels

For the next acoustic analysis, only half of the previously tested 3D-printed double panel samples (8 mm thick) were used to determine whether the thickness of the tested sample influenced the acoustic performance. Regarding the sound absorption coefficient (Figure 8a,c,e), the acoustic tests showed the same types of configurations, as with the double panels with the highest values (40% triangular–PLA–coconut, corrugated PLA–birch and 80% triangular–*PLA*). The increase in the absorption coefficient for the three types of samples, compared to the results obtained for the double panels, was as follows: 39% for the single 40% triangular–PLA–coconut samples; 20% for the single 100% corrugated PLA–birch samples and 33% for the single 80% triangular–*PLA* samples. Therefore, it can be stated that sample thickness is very important in acoustic tests, as previously found in other studies [40,54,63].

Regarding the sound transmission loss (Figure 8b,d,f), as found in other studies [53,64], the STL value decreases as the thickness of the tested samples decreases, and the maximum value is obtained at a higher frequency (in the case of these samples, at about 2500 Hz). Thus, the values of sound transmission loss for 3D-printed single panels decreased by about 13% compared to 3D-printed double panels. It can also be observed that the corrugated configuration shows the highest STL values for all three material types.

Figure 8 shows that the decrease in the thickness of the specimen, from 16 mm to 8 mm increased the ability to absorb sound at frequencies between 1000 Hz and 2500 Hz. As the thickness of the 3D-printed specimens increases, the acoustic absorption decreases due to the low and mid frequency because the waves have a harder time penetrating the thicker panels. If the thickness of the specimens is too large, the air inside the voids becomes harder to move, and there will effectively be no friction. It can be concluded that the sound absorption coefficient improved at lower frequencies (500 Hz) and, at the same time, an increase was observed when the frequencies were increased (3150 Hz). The decrease in the filling density in most of the tested specimens improves the passage of the acoustic waves to enter the analysed structure, and the air inside the voids can move easily, which provides an increase in the viscous friction, ultimately causing the loss of the energy of the acoustic waves, causing higher sound absorption [51]. By decreasing the thickness of the specimen (from 16 mm to 8 mm), the values of the sound transmission loss parameter decreased, but the specimens still offer good acoustic insulation with values between 30 dB and 55 dB.

### 3.5. Acoustic Analysis of 3D-Printed Single Panels with Drilled Holes

In these tests, the 3D-printed single panels were perforated to determine the influence of holes on acoustic performance. A template (Table 1) was used to make the holes, and the 3D-printed single panels were then perforated. The holes were round, made with a drill, and the diameter was 1.8 mm. They were made at right angles, as they have a higher absorption coefficient compared to the holes made at inclined angles [39]. After acoustic testing, 3D-printed single panels with holes showed superior performance compared to 3D-printed single panels. From the analysis of the acoustic test results, in comparison with the results obtained on 3D-printed single panels and according to the highest absorption coefficient values, the following was found: 40% triangular–PLA–coconut samples showed a 73% increase in absorption coefficient; 40% triangular–PLA–birch samples showed a 122% increase in absorption coefficient; 60% triangular–*PLA* samples showed an 82% increase in absorption coefficient. The triangular configuration also had the highest absorption coefficient values for all three material types. The filling densities which indicated the highest absorption coefficient (Figure 9a,c,e) were 40% and 60%. Another important aspect of the tests on 3D-printed single panels with holes was that with the use of the holes, the peak sound absorption coefficients shifted to the value of 1000 Hz [65]. Concerning the material, it can be concluded that PLA had the highest absorption coefficient of 0.86, while for the birch material, a maximum absorption coefficient value of 0.78 was obtained. Thus, it can be implied that the holes of the samples brought a significant increase in the sound absorption coefficient, as observed in another research conducted [29,39,40,66]. The absorption coefficient curves (Figure 9a,c,e) have stabilized, and their shape is similar for all material types, with very small variations between them.

Regarding the sound transmission loss of 3D-printed single panels with holes, it can be observed that the maximum value is found at 3150 Hz, so the sound wave energy dissipation is directly proportional to the frequency [29]. As can be seen from the curves of sound transmission loss (Figure 9b,d,f), they have a similar shape, with a maximum of about 29 dB. Thus, a decrease in sound transmission loss of up to half is observed compared to 3D-printed single panels, and the maximum value in this case was 29 dB for the 60% triangular–PLA–coconut sample.

### 3.6. Acoustic Analysis of 3D-Printed Single Panels with Holes

In the acoustic testing of 3D-printed single panels with drilled holes, the best performance (in terms of absorption coefficient) was recorded for the following samples: 40% triangular–PLA–coconut; 40% triangular–PLA–birch; 60% triangular–*PLA*. For these configurations, filling densities and material samples were manufactured via the FFF process, and the results obtained were similar.

Therefore, to optimise the structures, the three configurations (40% triangular–PLA–coconut; 40% triangular–PLA–birch and 60% triangular–*PLA*) were kept and samples with rhombic holes of 1.8 mm side size were manufactured via the FFF process (Table 1). From the absorption coefficient curves (Figure 10a) the maximum value is found for the 40% triangular–PLA–birch sample of 0.93 at a frequency of 1000 Hz. In contrast, for the 60% triangular–*PLA* sample, the maximum value of the absorption coefficient is 0.84 at 1600 Hz.

From this, it can be concluded that PLA with a mixture of 40% ground birch wood particles can be used at high performance at frequencies up to 1000 Hz and PLA material up to 1600 Hz for the configurations proposed in this study. The 40% triangular–PLA–birch sample with 3D-printed rhombic holes showed a 16% increase compared to the 40% triangular–PLA–birch with drilled holes sample in terms of absorption coefficient value. From the tests, it can be concluded that the triangular structure, at a filling density of 40% and with rhombic holes showed the highest absorption coefficient (0.93).

As for the sound transmission loss (Figure 10b), using rhombic shaped holes increased it by 10% compared to 3D-printed single panels with drilled holes. The maximum STL value is 31 dB and is attributed to the 60% triangular–*PLA* sample at a frequency of 3150 Hz. The shape of the curves for sound absorption coefficient and for sound transmission loss for 3D-printed single panels with holes shows the same shape as 3D-printed single panels with drilled holes.

Even though perforations of 3D-printed specimens can be seen to improve sound absorption, this is due to the inherent disadvantage demonstrated by micro-perforation in suppressing low-frequency (i.e., <1000 Hz) noise, as observed in other studies [53,67]. This aspect was also validated in the study in this work, namely that in the case of perforated panels, the peak values of the sound absorption coefficient were near the frequency of 1000 Hz (Figure 9 and Figure 10). On the other hand, open porous structures could dampen sound; this aspect is related to the airflow resistivity of these panels [31]. In general, increasing airflow resistivity improves sound absorption properties [31] over the entire frequency range, but only up to an intermediate value.

Because the research started from the analysis of demountable double panels and the optimal one turned out to be a half-panel, this study has comparable results with those obtained in other studies by other researchers.

## 4. Conclusions

Acoustic analysis and noise absorption are some important and research-intensive aspects of health and the environment that affect people’s psychological and biological states.

In this work, the sound absorption performance of samples manufactured from three types of materials (PLA with a mixture of 40% ground coconut wood particles, PLA with a mixture of 40% ground birch wood particles and plain PLA), with three internal configurations (circular, triangular, and corrugated) and at five filling densities (20%, 40%, 60%, 80% and 100%), manufactured via the FFF process, was evaluated.

The acoustic performance was measured for the double panels (designed as consisting of two half-panels in a demountable assembly) and obtained a maximum value of α = 0.4 and STL = 64 dB. The acoustic results for the double panels (16 mm thickness) indicate excellent sound transmission resistance for all types of materials used, which determines good acoustic insulation, and these types of materials can be successfully used in the aviation, automotive and construction fields for sandwich panels with the aim of sound attenuation.

After several tests, a compromise solution was reached that involved form iterations, panel perforations, and porosity optimization to increase the acoustic performance of the samples considerably, with values ranging from 72% to 122%.

For a complete characterization of the samples, mechanical tests were also carried out, and the PLA material was clearly superior to the other two types of materials (PLA with a mixture of 40% ground coconut wood particles, PLA with a mixture of 40% ground birch wood particles) in both compression and three-point bending tests.

At the mechanical tests, the rhombic holes that presented the best sound absorption results with α = 0.93 and STL = 31 dB also presented the maximum resistance to compression (40 MPa) and to three-point bending (50 MPa).

Microscopic analysis of the samples tested in compression showed normal defects (voids, interlayer voids) of the FFF process and lateral buckling as the main mode of failure of the internal walls of the samples.

The analysis of the microstructure of the panels confirms that it has been possible to produce samples with controlled morphology and oriented regular cell patterns that could be attained and with high sound-absorbing properties.

In the 3D-printed samples with rhombic holes, an increase of 16% in the absorption coefficient of the PLA material with a mixture of 40% ground birch wood particles (α = 0.93) was obtained compared to the samples with round holes.

Predominant factors that influenced sound-absorbing properties in this study were the triangular configuration model, which exhibited the highest performance for all three material types, and filling density, which also played an important role in the acoustic tests of the samples manufactured via the FFF process; thus, it was determined that the highest acoustic performance was obtained at 40% and 60%. The effects of cell orientation impact the acoustic properties as the un-oriented cell morphology leads to enhanced sound absorption capacity compared to the samples with more regular and oriented morphology.

Depending on the field of application, the following are recommended:-For very good sound absorption properties, the configuration with 40% infill, triangular–PLA with coconut ground particles and 8 mm thickness is recommended.-The physical–mechanical properties of the material (PLA with ground coconut particles) ensure their use for obtaining protective sound-absorbing panels (near highways or heavily trafficked roads, various casings or covers of engines).-Transmission loss coefficient (STL) decreases with panel thickness.

As a general observation, the initial idea to obtain highway sound protection panels with the highest possible mechanical resistance and very good sound absorption properties was achieved.

Even if the results obtained are very good from the point of view of the sound absorption coefficient (α = 0.93) and the mechanical resistances (40 MPa and 50 MPa), additional scientific research is considered to increase the transmission loss coefficient and at the same time to maintain the α coefficient.

## Figures and Tables

**Figure 1 polymers-15-03695-f001:**
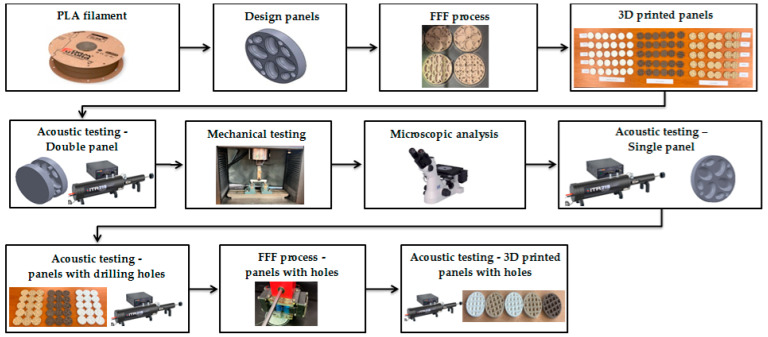
Flow chart of the present study.

**Figure 2 polymers-15-03695-f002:**
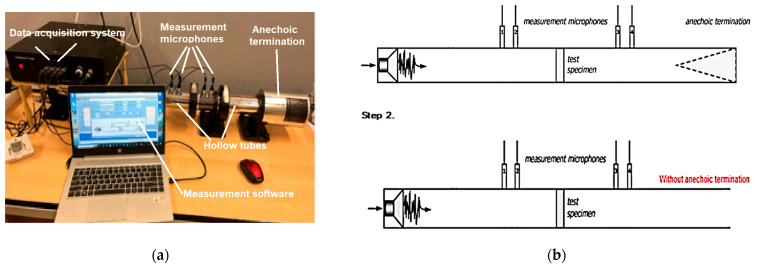
Experimental setup of the acoustic testing: (**a**) Equipment used for acoustic testing of samples manufactured via the FFF process; (**b**) Method of measurement of the sound absorption coefficient and of the sound transmission loss.

**Figure 3 polymers-15-03695-f003:**
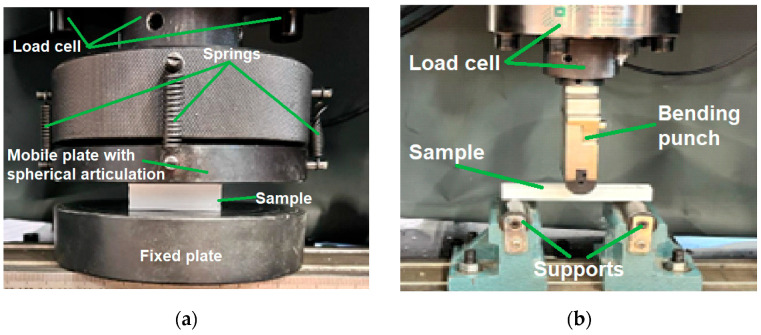
Mechanical testing: (**a**) Compression testing of the 3D-printed panels; (**b**) Three-point bending testing of the 3D-printed panels.

**Figure 4 polymers-15-03695-f004:**
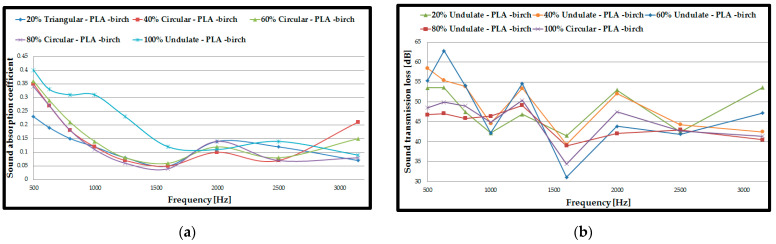
Acoustic test results of assembled double panels: (**a**) Sound absorption coefficient of PLA–birch samples; (**b**) Sound transmission loss of PLA–birch samples; (**c**) Sound absorption coefficient of PLA–coconut samples; (**d**) Sound transmission loss of PLA–coconut samples; (**e**) Sound absorption coefficient of PLA samples; (**f**) Sound transmission loss of PLA samples.

**Figure 5 polymers-15-03695-f005:**
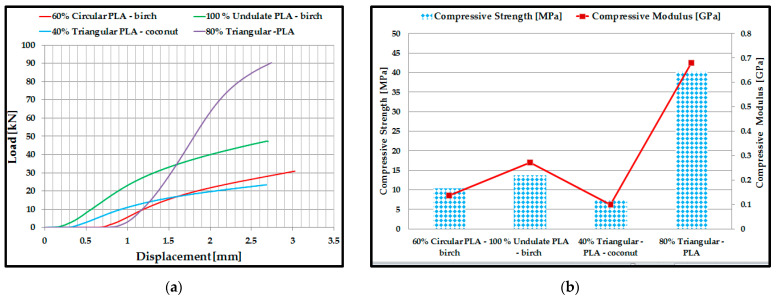
Compression test results: (**a**) Load–displacement characteristic curves; (**b**) Compressive strength and modulus of elasticity for 3D-printed samples.

**Figure 6 polymers-15-03695-f006:**
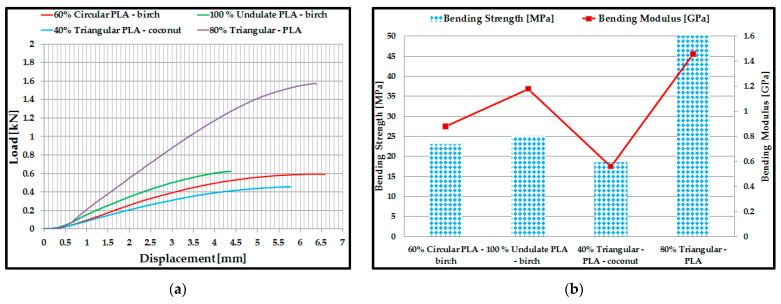
Three-point bending test results: (**a**) Load–displacement characteristic curves; (**b**) Bending strength and modulus of elasticity for 3D-printed samples.

**Figure 7 polymers-15-03695-f007:**
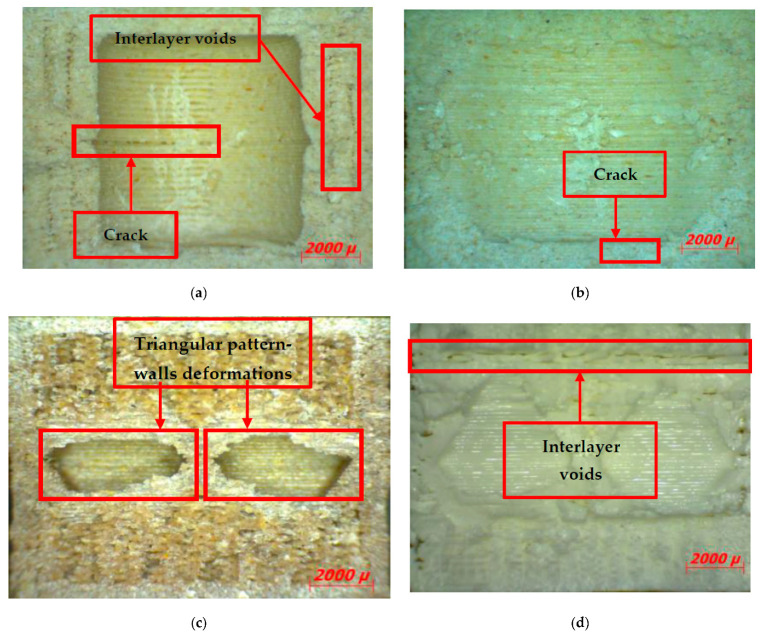
Microscopic analysis of the samples: (**a**) 60% circular PLA–birch; (**b**) 100% corrugated PLA–birch; (**c**) 40% triangular–PLA–coconut; (**d**) 80% triangular–PLA.

**Figure 8 polymers-15-03695-f008:**
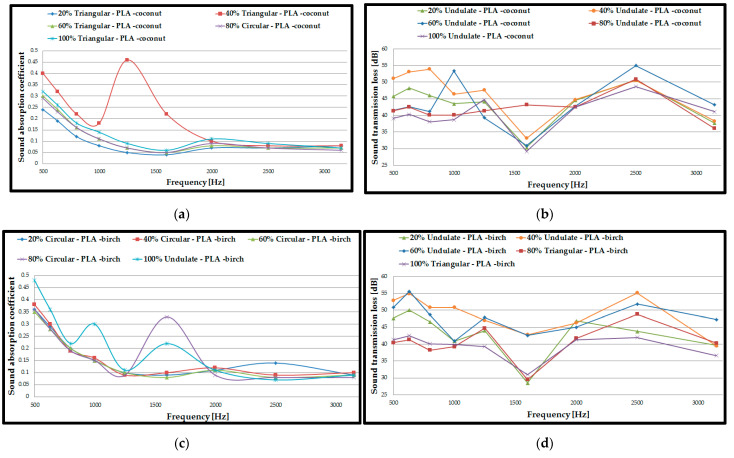
Acoustic test results of 3D-printed single panels: (**a**) Sound absorption coefficient of PLA–birch specimens; (**b**) Sound transmission loss of PLA–birch specimens; (**c**) Sound absorption coefficient of PLA–coconut specimens; (**d**) Sound transmission loss of PLA–coconut specimens; (**e**) Sound absorption coefficient of PLA specimens; (**f**) Sound transmission loss of PLA specimens.

**Figure 9 polymers-15-03695-f009:**
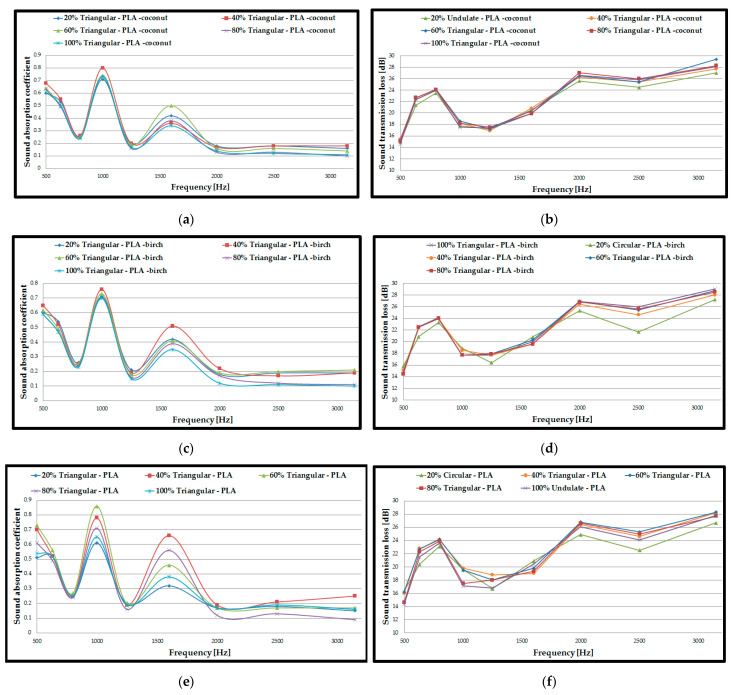
Acoustic tests results of 3D-printed single panels with drilled holes: (**a**) Sound absorption coefficient of PLA–birch specimens; (**b**) Sound transmission loss of PLA–birch specimens; (**c**) Sound absorption coefficient of PLA–coconut specimens; (**d**) Sound transmission loss of PLA–coconut specimens; (**e**) Sound absorption coefficient of PLA specimens; (**f**) Sound transmission loss of PLA specimens.

**Figure 10 polymers-15-03695-f010:**
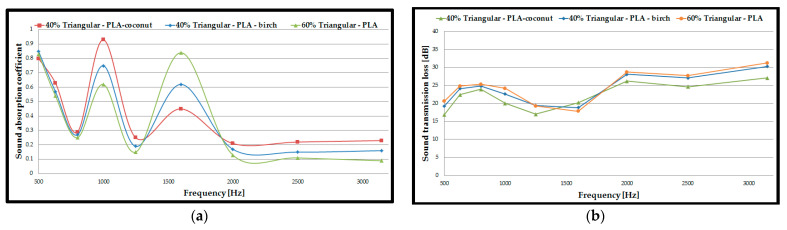
Acoustic test results of 3D-printed single panels with holes: (**a**) Sound absorption coefficient of 40% triangular–PLA–coconut; 40% triangular–PLA–birch and 60% triangular–PLA; (**b**) Sound transmission loss of 40% triangular–PLA–coconut; 40% triangular–PLA–birch and 60% triangular–PLA.

**Table 1 polymers-15-03695-t001:** The 3D-printed samples used for acoustic testing.

	Acoustic Test Specimens			
	Double panel	Single panel	Panels with drilling holes (Drill gage)	3D-printed panels with holes
Circular profile	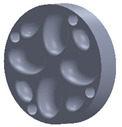 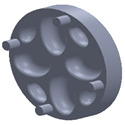	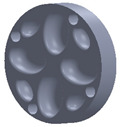	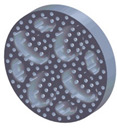	
Triangular profile	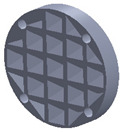 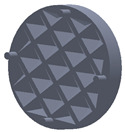	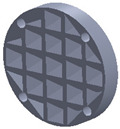	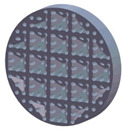	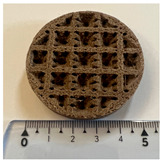
Corrugated profile	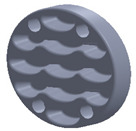 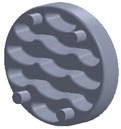	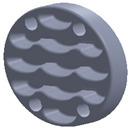	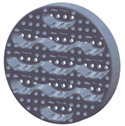	

**Table 2 polymers-15-03695-t002:** The 3D-printed samples used for mechanical testing.

	Compression Test Specimens		Three-Point Bending Test Specimens	
Circular profile	Section view	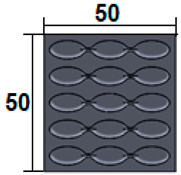	Section view	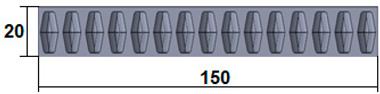
Triangular profile	Section view	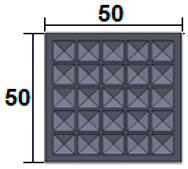	Section view	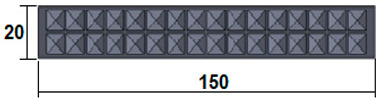
Corrugated profile	Section view	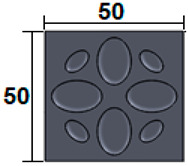	Section view	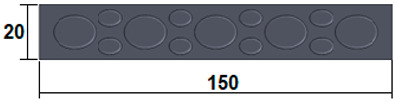

**Table 3 polymers-15-03695-t003:** FFF parameters of the 3D-printed panels.

Parameter	Value		
	PLA Coconut	PLA Birch	PLA
Filament diameter	2.85 [mm]	2.85 [mm]	2.85 [mm]
Layer height	0.2 [mm]	0.2 [mm]	0.2 [mm]
Infill density	20; 40; 60; 80; 100 [%]	20; 40; 60; 80; 100 [%]	20; 40; 60; 80; 100 [%]
Print speed	50 [mm/s]	50 [mm/s]	40 [mm/s]
Travel speed	120 [mm/s]	120 [mm/s]	120 [mm/s]
Printing temperature	240 [°C]	240 [°C]	240 [°C]
Building plate temperature	50 [°C]	50 [°C]	60 [°C]
Infill pattern	cubic	cubic	cubic
Hotend	0.4 [mm]	0.4 [mm]	0.4 [mm]

## Data Availability

The data presented in this study are available on request from the corresponding author.

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
