# Peer review of "Sound Absorption Performance and Mechanical Properties of the 3D-Printed Bio-Degradable Panels"

_polymers, 2023, doi:10.3390/polym15183695_

Round 1

Reviewer 1 Report

Sound Absorption Performance and Mechanical Properties of the 3D Printed Bio-Degradable Panels

Article is interesting. Few observations are given below;

Abstract need revision with some quantitative results.

Some more studies are required in the introduction section to further highlight the importance of this study.

Section 2.5 which standards were followed for testing.

Section 3.1, it is not necessary to explain such details again

"A major advantage of the FFF manufacturing process is that it can create parts with a lightweight structure by using and modifying the infill density parameter"

The authors must focus on results under these sections.

Section 3.2, results need present in better way.

As such, authors presented very limited results. There is no significant results in this study.

Authors must summarized results in more systematic way with reference to the previous studies.
Also, Conclusions are too limited to proof the significant outcome of this study.

Minor editing of English language required

Author Response

I greatly appreciate your careful review of our manuscript and the valuable suggestions you provided to enhance our paper. The revisions have been documented below. Our explicit responses addressing the necessary corrections are indicated in red writing and yellow highlight within the manuscript.

Reviewer 2 Report

The work presented by Zaharia et al. fabricated structural panels with three bio-degradable materials using 3D printing technique and investigated their mechanical and acoustic performance. The designed corrugated profiles is interesting and useful. The experimental results are of referential value to some degree for the researchers in the acoustic simulation field. Before it can be accepted for publication in Polymers, there are some minor concerns need to be addressed carefully by the authors.

1. The insets in Figure 1 are too small to recognize. Please rearrange them.

2. Three profiles were selected for 3D printed samples. Please provide explanation for why these profiles were selected and what is the design criteria.

3. Please add scale bars in Table 1 and 2 to demonstrate the real sizes of the panels.

4. Necessary annotations should be added in Figure 2(a) and Figure 3 to show the main subassembly of the testing setups.

5. The scale bars in Figure 7 are not normative. Please correct.

Minor editing of English language required

Author Response

(The authors gave the same response as above.)

Reviewer 3 Report

Sound Absorption Performance and Mechanical Properties of 3D Printed Bio-Degradable Panels 

General 

This submission focuses on the results of the determination of some acoustic and mechanical properties of 3D printed material layers of varying printing geometry and varying composition of biodegradable material. The whole exercise is quite elaborate: it involves an extensive measurement campaign as reported in the text.

Since the title refers explicitly to acoustic properties and since this is the reviewer's area of expertise, the comments relate mainly to these.

The overall assessment of this reviewer is: the work leaves too much to be desired on the topic of acoustics and is therefore not directly suitable for publication in an international scientific journal. Essential insights regarding the absorbing mechanisms in porous materials are missing.

In terms of the measurement technique used and the expression of the results, for example, Jung, S.S., Kim, Y.T., and Lee, Y.B. 2008. Measurement of sound transmission loss by using impedance tubes. Journal of the Korean Physical Society. 53: 596-600, could be a good starting point for an improved version.

In terms of the physical mechanism of sound absorption in porous materials and the determining material parameters, the authors can draw inspiration from CHAPTER 5 Sound absorbers in Building Acoustics, Tor Erik Vigran, Taylor & Francis (London and New York) (2008), ISBN: 9780415428538.

A thorough reworking and subsequent new review procedure appear necessary.

Comments and suggestions (not exhaustive)

19 It should be clear what infill is involved.

21 The sound absorption performance should be assessed in a way relevant to acoustics!

22 The size of the granules must be specified in the text body

29 It has by no means become clear why, how, or where one could deploy these 3D printed materials at an acceptable cost...

49 "When sound reaches a barrier, depending on the sound absorption performance of the material, three phenomena can occur: absorption, transmission, and reflection [18]" Depending on the nature of the material, we can expect other mechanisms to occur... So the material needs to be better specified.

54 57 "ratio of absorbed energy to incident energy " The proper physical quantity is sound intensity (W/m2).

61 "From the reviewed studies [23-27], the most important factor influencing the acoustic performance of materials is the porosity value. " This is indeed a prerequisite: open porosity, but why the test materials have not been investigated on this point? Besides, several additional characteristics are decisive: thickness, density, airflow resistance, structure factor, and thermal length (see e.g. reference Vigran)!

63 "…may replace… " This is undoubtedly a strong statement; it seems to me more realistic to mention that it could play a role.

107 "... to have good sound absorption performance, they were tested at high frequencies from 4800Hz to 6400Hz. " It is unclear which objectives and criteria are used for the assessment good.

Table 1 -2 It should be made clear somewhere in the text (211) why double panels are used. A 3D representation of a 3D-printed panel would be so much clearer!

157 "PLA reinforced 157 with 40% grained wood particle " Is reinforced appropriate in this case?

Figure 4 It is unusual to display acoustic results on a linear frequency scale! Furthermore, STL is typically expressed in dB because of its link with auditory perception. 

337 "Therefore, it can be stated that sample thickness is very important in acoustic tests, as previously obtained in other studies" This is self-evident, it should be a starting point, not a conclusion of an experiment....

Looks fine.

Author Response

(The authors gave the same response as above.)

Round 2

Reviewer 3 Report

+ 56 When sound reaches a barrier, depending ... but not limited to 

Change to:

The sound absorption reflects the ratio between reflected and incident sound intensity (W/m2). 

+ At an appropriate place, the text should mention testing by use of plane running waves incident perpendicular to the tested surface (tube method). This is in contrast to a test with omnidirectional sound incidence in the reverberation chamber.

+ In response to my comments regarding the meaning and units commonly used in the acoustic field (dB), the authors have misunderstood the message: they have simply multiplied the relevant measurement results by 100! The numbers given are therefore meaninglessIn addition, where the absorption coefficient increases, the transmission factor must decrease and internal loss also plays a part in this. In principle, both added together the sum should not exceed 1.

To avoid new errors, I suggest changing the designation to transmission factor instead of sound transmission loss. This should also be adjusted in the figures.

Conclusion and advice: 

Interesting improvements and additions but there is an error that needs to be fixed. Suggestions were given.

Author Response

I greatly appreciate your careful review of our manuscript and the valuable suggestions you provided to enhance our paper. The revisions have been documented below. Our explicit responses addressing the necessary corrections are indicated in text with track changes.
